# Environmental Efficiency Assessment of Heavy Pollution Industry by Data Envelopment Analysis and Malmquist Index Analysis: Empirical Evidence from China

**DOI:** 10.3390/ijerph18115761

**Published:** 2021-05-27

**Authors:** Jun Xu, Yuchen Jiang, Xin Guo, Li Jiang

**Affiliations:** School of Business, Jiangsu Normal University, Xuzhou 221116, China; lmmxjxj@163.com (J.X.); 18281518350@163.com (X.G.); jlcl1003@163.com (L.J.)

**Keywords:** environmental efficiency, heavy pollution industry, DEA, Malmquist index, environmental pollution, global problem

## Abstract

Industrial waste discharged by heavy pollution industry is one of the main causes of global environmental degradation. Research on the environmental efficiency of high-polluting industry is necessary to tackle the problem of global environmental pollution. Using panel data of 19 sub-industries in China’s heavy pollution industry from 2001 to 2015, this article employs Data Envelopment Analysis (DEA) and Malmquist index (MI) to measure the environmental efficiency of heavy pollution industry from both the dynamic and static perspectives. The results show that the environmental efficiency of China’s heavy pollution industry maintains an upward trend but did not reach the optimal level. The general trend shows a phased trend of increasing first and then decreasing. Besides, there are inter-industry differences in the environmental efficiency across the examined sub-industries. Based on the research findings, this article proposes a set of corresponding countermeasures to solve the global pollution problem, such as reducing energy inputs and minimizing the volumes of the main categories of emissions in high-polluting industry, as well as improving the production management in the group of high environmental efficiency and strengthening technical capabilities in the group of low environmental efficiency.

## 1. Introduction

Due to economic globalization, the economies of most countries in the world are constantly developing. However, the serious consequence of that process is environmental pollution. Many scholars have studied the relationship between environmental pollution and economic growth. For instance, scholars verified the validity of the Environmental Kuznets Curve (EKC) [1,2,3]. In other words, when the level of economic development is low, the degree of environmental pollution is low as well. As the economy becomes more developed, the degree of pollution gradually increases. Then, after the economic development reaches a certain level, the degree of pollution begins to decrease. On the other hand, environmental pollution has a detrimental effect on economic development [4]. As environmental pollution becomes more and more serious, its negative impact on economic development also increases. At present, national economic progress is inseparable from industrial development, especially in developing countries. For example, China’s economic growth is mainly attributed to the industrial sector. However, industrial sectors also have a great impact on the environment [5]. In this context, research on the pollution caused by high-polluting industries plays a crucial role in the fight against global environmental degradation and should not be underestimated. How to evaluate the impact of high-polluting industries on the environment? On the basis of evaluating its pollutants, it should also take into account the impact of its inputs and other outputs. By measuring environmental efficiency, the impact on the environment should be comprehensively measured.

For the definition of environmental efficiency, this article adopts the definition proposed by Li and Luo (2016) “Environmental efficiency means that a certain area uses less environmental resources and produces less ecological impact, so as to provide products and services that meet human needs” [6]. In recent years, scholars have become more and more interested in measuring the environmental efficiency of various regions, countries, or economies. The environmental efficiency is developed from ecological efficiency [7]. The evaluation of environmental efficiency can solve the problem of incomplete utilization of resources in the industry from the perspective of input and output, which leads to pollution and waste. At present, there is still a lack of empirical research on the environmental efficiency measurement of the industry. As the most polluting industry, studying its environmental efficiency will help us find countermeasures in the field of high-pollution industries, improve the efficiency of resource utilization in high-pollution industries, reduce pollution, and treat one of the important sources of pollution.

There are numerous methods of evaluating environmental efficiency, including life cycle approach, balanced scoring method, analytic hierarchy process and fuzzy comprehensive evaluation method. Later, the research community saw the emergence and development of Data Envelopment Analysis (DEA). DEA is increasingly applied in the evaluation of environmental efficiency [8]. When evaluating the effectiveness of a complex system with multiple inputs and multiple outputs, the DEA model can directly enter input and output data to establish an optimal model, which is more comprehensive than other methods. The continuous development of DEA theory has further improved its shortcomings. Applying the DEA model to measuring environmental efficiency makes a significant contribution to our comprehensive and objective understanding of high-polluting industries. Therefore, this article uses the DEA model to study heavy pollution industry. The biggest feature that distinguishes a heavy pollution industry from other industries is that it produces dramatically more pollutants compared to other industries. Thus, when measuring its environmental efficiency, we should fully consider the issue of pollutant emissions. More specifically, it is necessary to adopt a DEA model that includes undesirable output to measure the environmental efficiency of high-polluting industry. Additionally, from a dynamic perspective, the MI is used to measure the change of environmental efficiency, and then, corresponding countermeasures and suggestions are put forward. Environmental pollution is a major global problem, which is especially acute in developing countries.

Therefore, this article selects China as the research object of this article. China is a representative developing economy or one of the most representative developing economies in the world. Besides, some data about production in China would make the need for such analysis easier to present and they might be helpful in the context of results and conclusions. This paper studies the environmental efficiency of China’s heavy pollution industry, with a view to putting forward policies and suggestions on environmental pollution problems in China and the world.

The main purpose of this article is to measure and analyze the environmental efficiency of high-polluting industries as a whole. Research on the high-polluting industries as a whole is conducive to proposing targeted measures to improve environmental efficiency, which is different from other industries. At the same time, we try to study the differences between sub-sectors of high-polluting industries, so that different polluting industries can take different measures according to their industry characteristics to solve pollution problems.

The second part of this article is to sort out and analyze the literature related to heavy pollution industry and environmental efficiency. The third part is to briefly introduce the methods used in this article, including DEA model with undesirable output and MI. In addition, not only does this part explain the source of the data, but it also determines the research object. The fourth part is the data analysis and results. The last part summarizes the conclusions of this research, and then puts forward corresponding policy recommendations based on the research findings.

## 2. Literature Review

The literature examined in this article mainly concerns heavy pollution industry, environmental efficiency and methods used to measure environmental efficiency.

Based on previous studies, this article finds that as the pollution problem becomes more severe, people increasingly focus on the measurement of environmental efficiency in different countries and regions. Park et al. (2018) conducted a study on the transportation industry of 50 states in the United States from 2004 to 2012 and measured their environmental efficiency [9]. Capital, energy, and labor were selected as input variables. The transportation value added was the desirable output, and the CO_2_ emissions were the undesirable output. The results of the study showed that the average environmental efficiency score of each state in the United States was less than 0.64. Le et al. (2019) conducted a study on agricultural production efficiency and environmental efficiency in nine Southeast Asian countries from 2002 to 2010 [10]. The article used capital, labor, agricultural land, and fertilizer application as input indicators, and gross agricultural output value as desirable output. Wei et al. (2020) studied the environmental efficiency of countries along the “Belt and Road” from 1990 to 2014 and compared it with countries not along the “Belt and Road” [11]. In addition, this article selects total labor, total capital, and total energy consumption as input variables, GDP as desirable input, and carbon dioxide as undesirable input. Iram et al. (2020) evaluated the energy and economic environmental efficiency of the OECD countries from 2013 to 2017, using primary energy consumption and population as input variables, GDP as desirable output, and carbon dioxide emissions as undesirable output [8]. Hermoso-Orzáez et al. (2020) measured the environmental efficiency of the 28 EU member states from 2005 to 2012 [7]. The indicators in this article are different from those selected in the previous articles. Coal power generation, petroleum power generation, nuclear power generation, industrial production index, and traffic volume are used as input indicators. Desirable output also selected rent per capital, and undesirable output is CH4 emissions. Fukuyama et al. (2020) measured Japan’s environmental efficiency by the two-stage network production model in their research and studied the relationship between environmental and energy efficiency and Japanese happiness [12]. Twum et al. (2021) measured the environmental efficiency values of different regions in the Asia-Pacific region from 1990 to 2018 by super-efficiency DEA model and proved the inverted U-shaped link between environmental efficiency and technological innovation [13]. An et al. (2020) conducted a study on China’s regional industries from 2011 to 2015, regarding SO_2_ and solid waste emissions as undesirable output [14]. The study showed that there are significant differences in environmental efficiency between China’s developed regions and developing regions, and the gap is increasing year by year. There are many studies on the environmental efficiency of regions and countries, but there is a lack of research on the environmental efficiency of the industrial sector. Thus, this paper is different from most articles, as it selects industry as the research object.

In a small number of studies on industrial environmental efficiency and pollution problems, scholars often focus on a specific heavy pollution industry, studying the pollution and other problems it causes to the environment. Seklaoui et al. (2016) selected the mining industry and conducted a scientific assessment of the environmental pollution caused by it [15]. Hutton and Shafahi (2019) selected the leather industry and studied the waste of water pollution in the leather manufacturing process [16]. Zhang et al. (2020) selected the chemical industry and studied the impact of the air pollution prevention and control action plan on its green development [17]. Sun (2020) conducted a research on the power industry, measured the environmental efficiency of China’s power industry and studied the impact of market segmentation on it [18]. Wang et al. (2020) studied the steel industry, measured the environmental efficiency of China’s steel industry, and proved that there are regional efficiency differences [19]. Li and Xiao (2020) selected the paper industry as the research object and studied the environmental efficiency of the U.S. paper industry from 2015 to 2018 and measured its average efficiency value of 0.509 [20]. Although research on specific industries is conducive to putting forward specific solutions, the lack of comparison with other heavy pollution industries makes it difficult to understand the heterogeneity of industrial environmental efficiency. Therefore, this article takes the overall heavy pollution industry as the research object, measures its environmental efficiency, and identifies the differences in sub-industries through the comparison between them. At the same time, suggestions are made for the general environmental improvement of heavy pollution industry.

For the measurement of environmental efficiency, different scholars have adopted different methods, but they are roughly divided into two categories. One is the relatively simple single ratio method, which Sun (2020) elaborated in its overview of the development of environmental efficiency at home and abroad, and the other is the DEA method [18]. With the continuous enrichment of DEA theory, the DEA method of measuring environmental efficiency is constantly being improved. Chen et al. (2018) constructed a non-cooperative and cooperative DEA method to evaluate the environmental efficiency of decision-making units (DMUs) [21]. Wang et al. (2018) combined DEA with the material balance principle of the law of thermodynamics when measuring the environmental efficiency of the thermal power industry [22]. Shah and Longsheng (2020) developed a slack-based environmental efficiency index to study the environmental efficiency of Pakistan’s main economic sectors [23]. Singpai and Wu (2021) combined the two-stage DEA method with an extended Kaya identity-based Logarithmic Mean Divisia Index (LMDI) decomposition analysis when measuring environmental efficiency [24]. Most of the environmental efficiency measurement methods use DEA. Hu and Lu (2018) studied the impact of non-dimensional data and the correlation between indicators on the evaluation results of the DEA model [25]. This study proved that the dimensionlessness of the data does not affect the evaluation results of the DEA model. The DEA model can effectively evaluate complex systems with multiple inputs and multiple outputs. At the same time, the key to distinguishing heavy pollution industry from other industries is “pollution”. The emissions cannot be ignored. Therefore, this article uses a DEA model that includes undesirable output, which can effectively analyze the environmental efficiency of heavy pollution industry. We adopt the DEA method to evaluate the environmental efficiency of the industry from a static perspective. In addition, MI is applied to measure the changes in environmental efficiency from a dynamic perspective. By taking both the static and dynamic approaches, we conduct a comprehensive analysis of the environmental efficiency of high-pollution industry, so as to propose the corresponding policies and suggestions for managing heavy pollution industries.

## 3. Empirical Design

This article adopts the DEA model that includes undesirable output to measure the environmental efficiency of 19 sub-industries of China’s high pollution industry from 2001 to 2015. In order to evaluate the environmental efficiency level of China’s heavy pollution industry comprehensively, this article uses MI to further analyze the change in the environmental efficiency of China’s heavy pollution industry.

### 3.1. Method

#### 3.1.1. DEA Model with Undesirable Output

In order to evaluate non-parametric efficiency, Charnes, Cooper and Rhodes first proposed DEA [26], which is an effective method for estimating the production frontier. The production process creates not only the products we need but also some products that we do not need, such as carbon dioxide, sulfur dioxide, and solid wastes emitted during the production process. These unwanted by-products accompanying the production process are regarded as undesirable output. The traditional DEA theory does not take into account the problem of undesirable output. In existing studies, measures to improve efficiency can reduce input or expend output, but expanded output also includes undesired output. Therefore, we need to adopt methods to deal with undesirable output. In order to overcome the shortcomings related to the fact that the traditional DEA model ignores the undesirable output, many scholars have adopted different methods to deal with the undesirable output in the production process. Since environmental pollution is the cost of economic activity, the less undesirable output, the better, it seems more intuitive to use undesirable output as input [27]. For instance, Zhou and Wu (2020) used the super-efficiency DEA model to measure the environmental efficiency of China’s advanced manufacturing industry, using wastewater, waste gas, and solid waste emissions as environmental inputs to measure [28]. In addition, there are methods such as the reciprocal method [27], hyperbolic processing method [29], and linear transformation method [30]. However, the method of using undesirable output as input and the reciprocal method violate the essence of production, while the hyperbolic method and the linear transformation method are computationally demanding [30,31].

At present, the most commonly used methods to deal with undesirable output are the directional distance function and the slack based model (SBM) [32]. The directional distance function is essentially a radial DEA, and it cannot deal with the problem of slack well. There is a big debate on the choice of the directional vector. Therefore, this article mainly adopts the SBM method, an improved DEA method.

Consider a production possibility set that contains bad output
(1)$$P=\left\{\left(x,y^{g},y^{b}\right)|x\geq X\lambda ,y^{g}\leq Y^{g}\lambda ,y^{b}\geq Y^{b}\lambda ,L\leq e\lambda \leq U,\lambda \geq 0\right\}$$

*x* is the input, *y^g^* is the desirable output, *y^b^* is the undesirable output, and *λ* is a weight vector.

The expression of the SBM model including undesirable output is
(2)$$\left\{ \begin{array}{l} \rho *=\min\frac{1-\frac{1}{m}\sum_{i=1}^{m}\frac{s_{io}^{-}}{x_{io}}}{1+\frac{1}{s}\left(\sum_{r=1}^{s_{1}}\frac{s_{r}^{g}}{y_{ro}^{g}}+\sum_{r=1}^{s_{2}}\frac{s_{r}^{b}}{y_{ro}^{b}}\right)}\\ x_{0}=X\lambda +s^{-}\\ y_{o}^{g}=Y\lambda -s^{g}\\ y_{o}^{b}=Y\lambda +s^{b}\\ L\leq e\lambda \leq U\\ s^{-},s^{g},s^{b},\lambda \geq 0 \end{array}\right.$$

$x_{io},\;y_{{}_{ro}}^{g},\;y_{ro}^{b}$ are the amount of input, desirable output, and undesirable output; $s,\;s_{1},\;s_{2}$ are the total number of outputs, the number of desirable outputs, and the number of undesirable outputs. DMU is most effective if and only if
$$\rho^{*}=1,\;i.e.,\;s^{-*}=0,s^{g*=0},s^{b*}=0$$

#### 3.1.2. Malmquist Index

The MI was first proposed by Sten Malmquist in 1953 [33]. It pays close attention to the dynamic changes in continuous time efficiency and productivity of similar DMUs. The MI can be decomposed into two parts: catch-up effect and frontier-shift effect Catch-up effect is the change effect of DMU efficiency, it reflects the change of DMU’s efforts to improve efficiency, and the improvement of management ability, which is also called the efficiency change (EFF). The frontier-shift effect reflects the shift of the production frontier referenced by all DMUs in the two periods, which is called the technical change (TECH).
(3)MI=EFF×TECH
(4)$$\mathrm{EFF}=\frac{\delta^{2}{\left(x_{o},y_{o}\right)}^{2}}{\delta^{1}{\left(x_{o},y_{o}\right)}^{1}}$$

*δ*^1^ (*x_o_*,*y_o_*)^1^, *δ*^2^ (*x_o_*,*y_o_*)^2^ respectively represents the efficiency score of the input and output in the first period corresponding to the frontier of the first period, and the input and output of the second period corresponding to the efficiency score of the frontier in the second period.
(5)TECH=[δ1((xo,yo)1)δ2((xo,yo)1)×δ1((xo,yo)2)δ2((xo,yo)2)]12

*δ*^2^ ((*x_o_*,*y_o_*)^1^) × *δ*^2^((*x_o_*,*y_o_*)^1^) respectively represents the efficiency value of input-output in period 1 relative to the frontier of period 2 and the efficiency value of input-output in period 2 relative to the frontier of period 1.

Therefore, when MI ≥ 1, the environmental efficiency of the industry is increasing, when MI < 1, it means that the environmental efficiency of the industry is decreasing; when TECH ≥ 1, the frontier movement is beneficial to efficiency, and when TECH < 1, the frontier movement is not good for efficiency; when EFF ≥ 1 means that the efficiency of a certain industry has improved compared with the previous year, when EFF < 1, the opposite is true.

### 3.2. Index Selection

Regarding the classification of heavy pollution industry, this article combines the 16 high-polluting industries specified in the Guidelines for Environmental Information Disclosure of Listed Companies issued by the Ministry of Environmental Protection in 2010 with the classification standards of Chinese national economic industries in the National Standards of the People’s Republic of China. A total of 19 heavy pollution sub-industries have been identified, namely, Mining and Washing of Coal (B06); Extraction of Petroleum and Natural Gas (B07); Mining and Processing of Ferrous Metal Ores (B08); Mining and Processing of Non-Ferrous Metal Ores (B09); Manufacture of Liquor, Beverages and Refined Tea (C15); Manufacture of Textile (C17); Manufacture of Textile, Wearing Apparel and Accessories (C18); Manufacture of Leather, Fur, Feather and Related Products and Footwear (C19); Manufacture of Paper and Paper Products (C22); Processing of Petroleum, Coking and Processing of Nuclear Fuel (C25); Manufacture of Raw Chemical Materials and Chemical Products (C26); Manufacture of Medicines (C27); Manufacture of Chemical Fibres (C28); Manufacture of Rubber and Plastics Products (C29); Manufacture of Non-metallic Mineral Products (C30); Smelting and Pressing of Ferrous Metals (C31); Smelting and Pressing of Non-ferrous Metals (C32); Manufacture of Metal Products (C33), and Production and Supply of Electric Power and Heat Power (D44).

In order to evaluate the environmental efficiency of the above 19 heavy pollution sub-industries comprehensively and to find the key factors that affect the sustainable development of the heavy pollution industry, this article focuses on the representativeness and quantification when selecting indicators. Combining the existing research on environmental efficiency and taking into account the availability of data related to the research objects of this article, the input and output indicators of this article are as follows.

Input indexes. Input indexes are mainly divided into three categories: energy input, labor input and capital input. This article refers to the indicators used by Sheng (2012) to measure environmental efficiency when studying the relationship between industrial environmental regulations and environmental efficiency [34]. The comprehensive energy consumption of each industry is selected to measure the energy input of heavy pollution industries, the annual average number of employees in each industry is used to measure labor input, and the net fixed assets of each industry is used to measure the capital input.

Output indexes. The output indicators mainly include desirable output and undesirable output. This article refers to the research of Liu and Zhang (2012), taking the main business income of each industry as the desirable output [35]. According to the practice of most scholars, the amount of wastewater discharge, exhaust gas discharge, and solid waste discharge are regarded as undesirable output. The “exhaust gas emissions” refer to “the various emissions generated during the fuel combustion and production processes in the company’s plant area during the reporting period. The total amount of pollutant gases in the air is calculated based on the standard state (273 K, 10,135 Pa)”.

### 3.3. Data

The research objects of this article are 19 sub-industries of China’s heavy pollution industry from 2001 to 2015. The data sources are as follows: the comprehensive energy consumption of each industry comes from the China Energy Statistical Yearbook, the number of employees in each industry and the main business income comes from the *China* Industry Statistical Yearbook and China Statistical Yearbook, and the data of wastewater discharge, exhaust gas discharge, and solid waste discharge in various industries come from the China Statistical Yearbook and China Statistics Yearbook on Environment.

## 4. Results and Analysis

### 4.1. Descriptive Statistics and Correlation Analysis

The descriptive statistics of various indicators in the environmental efficiency evaluation model of high-polluting industry are shown in Table 1. In order to better present the status quo of China’s high-pollution industries and facilitate subsequent analysis, this paper presents the main data of the indicators of 19 high-pollution sub-industries in Table 2.

Analyze the correlation between the indicators of the environmental efficiency model of high-polluting industries. This paper finds that there are significant correlations between energy input and capital input, between desirable output and energy input, labor input, and capital input, and between exhaust emissions and energy input, capital input, and main business income. There is no significant correlation between other variables.

### 4.2. Static Analysis of Environmental Efficiency Changes in Heavy Pollution Industries

#### 4.2.1. Overall Analysis of Environmental Efficiency in Heavy Pollution Industries

This article adopts the DEA model with undesirable output on the environmental efficiency of 19 sub-industries of China’s heavy pollution industry and uses DEA-Solver-Pro 13.1 (distributed by Beijing Huan Zhong Rui Chi Technology Co., Ltd. Beijing, China) to calculate the environmental efficiency values of China’s heavy pollution industry.

On the whole, the environmental efficiency of China’s heavy pollution industry demonstrated an upward trend from 2001 to 2015, but the growth rate was not high (Figure 1). It increased from 0.638 in 2001 to 0.826 in 2015, an increase of 29.47%. During the 15-year period, except for 2004, 2012, and 2014, which were lower than the previous year, the remaining years all saw positive growth, especially in 2003 and 2011, which increased by 20.52% and 9.61%, respectively. According to data, in 2003, China carried out inspections and enforcement of the Law of the People’s Republic of China on the Prevention and Control of Solid Wastes Pollution. A series of documents including the Regulations on the Management of Medical Waste and the National Planning for the Construction of Hazardous Waste and Medical Waste Disposal Facilities were issued. At the same time, the comprehensive utilization of straw, the management of imported waste, and the environmental management of the imports and exports of chemicals were strengthened nationwide, resulting in a decrease of 26.3% of solid waste emissions nationwide compared with last year, which greatly improved the environmental efficiency in 2003. In 2011, China’s environmental efficiency grew relatively high and reached the peak value. On the one hand, that can be explained by the low amount of undesirable output, that is, the reduction in solid waste emissions. The main reason was that China successively issued the Measures for the Administration of Solid Waste Imports, the Notices of the Ministry of Environmental Protection, the General Administration of Customs, and the General Administration of Quality Supervision, Inspection and Quarantine on Strengthening the Administration of Solid Waste Imports and the Sharing of Law Enforcement Information, etc., formally established China’s solid waste import management and law enforcement information communication and sharing mechanisms, reducing solid waste emissions. On the other hand, that growth in efficiency may also be attributed to the increase in investment, such as the increase in capital investment in Production and Supply of Electric Power and Heat Power (D44). Although the environmental efficiency of China’s heavy pollution industry has improved over the past 15 years, it has not reached the optimal efficiency value, and the overall level of China’s environmental efficiency has not reached the effective level.

#### 4.2.2. Environmental Efficiency Analysis of Heavy Pollution Sub-Industries

This article calculates the average environmental efficiency of China’s heavy pollution industry from 2001 to 2015 through the environmental efficiency values of 19 heavy pollution sub-industries and finds that the environmental efficiency of China’s high-polluting industry has obvious industry differences (Table 3).

Among the 19 industries, B07, C18, C19, C25, C27, C29, and C33 have the best efficiency. B07 ranks high in terms of fixed asset input, while the emissions of three wastes have always been in the bottom five. C18 energy input is less, and the emissions of three wastes are in the bottom three, so the efficiency reaches the best level. C19 energy input is the bottom one, fixed assets is the second bottom, and the emissions of three wastes are also at the bottom. The desirable output of C25 is higher, which leads to the best efficiency. C27 has less investment, but the amount of solid waste discharge is lower. The discharge of wastewater of C29 is the second to last, which improves the efficiency of resource utilization. The desirable of C33 ranks relatively high, and the discharge of wastewater and solid waste is relatively low. The industries with lower environmental efficiency are D44, C26, C31, C22, C30, and B06. D44 ranks fourth in energy input and first in fixed asset investment. Although its main business is also relatively high, it also ranks first in waste gas emissions. C26 ranks first in energy input and wastewater discharge, and its resource utilization rate is very low. C31 ranks first in energy input and second in capital investment, but the emissions of the three wastes are also large, and resources are not used effectively. The investment of C22 is moderate, but its wastewater discharge volume ranks first, resulting in low environmental efficiency. The wastewater discharge volume of C30 is actually not high, but its waste gas and solid waste discharge volume is relatively large, ranking in the top five. B06 has high investment in human and financial resources, but its main business income is low, and its solid waste emissions rank first, resulting in its lowest environmental efficiency. For other sub-industries, B08 ranks third in solid waste emissions, B09 ranks second in solid waste emissions, C15 has low input and output and high wastewater emissions, and C17 has high labor input and ranked third in wastewater emissions. The main business income of C28 is third from the bottom, and the exhaust emissions of C32 is fifth. All these have led to low environmental efficiency.

The environmental efficiency value of B06 reached 1 in 2011 because its main business income increased by 33.05% compared to 2010. The reason why the environmental efficiency value of B08 reached 1 in 2010 was because the investment in assets increased by 66.40% compared to 2009. The environmental efficiency value of B09 in 2009 is due to a reduction of 42.28% in waste gas emissions compared to 2008. The reason why the environmental efficiency value of C17 fluctuates greatly from 2011 to 2013 is because the fluctuating value of exhaust gas emissions is relatively large. The fluctuation of C28 in 2003 was due to the decrease of solid waste discharge and the increase of main business income. In 2013, C30′s main business revenue increased by 18.14%, exhaust gas emissions decreased by 57.12%, and solid waste emissions decreased by 70.51%. C31′s main business income increased by 30.71% in 2008. C32′s solid waste discharge in 2003 was only 20,000 tons. In 2011, D44 had less labor input and only 45,100 tons of solid waste discharge, which led to an environmental efficiency value of 1.

Figure 2 shows that the difference between the maximum and minimum environmental efficiency of the 19 sub-industries is 0.64. The reasons for such a large gap are as follows: Firstly, the three main categories of emissions in the industries with high environmental efficiency values are relatively low in volume. For example, the emissions of waste gas and solid waste from the Manufacture of Textile, Wearing Apparel and Accessories (C18) are the lowest among all industries, accounting for 0.05% and 0.06 of the entire polluting industry, respectively; The Extraction of Petroleum and Natural Gas (B07) has the lowest wastewater discharge, accounting for 0.62% of the entire pollution industry. Secondly, industries with low environmental efficiency have higher energy input, such as the Smelting and Pressing of Ferrous Metals (C31). Energy input is as high as 479.11 million tons of standard coal, accounting for 25.94% of all heavy pollution industries. Thirdly, industries with low environmental efficiency have higher capital investment, but their main business income is very low, such as the Production and Supply of Electric Power and Heat Power (D44). The capital investment is as high as CNY 3,897.96 million, accounting for 30.92% of all heavy pollution industries, but desirable output only accounts for 10.215%. Therefore, low energy consumption, high revenue, and low pollution are the keys to improving the environmental efficiency of heavy pollution industry.

According to the 15-year environmental efficiency value, the high-pollution industries in China are divided into three groups: high, medium, and low, from high to low (Table 4).

### 4.3. Dynamic Analysis of Environmental Efficiency Changes in Heavy Pollution Industry

To further study the changes in environmental efficiency of 19 high-polluting sub-industries in China, this article also uses solverpro13.0 to calculate the MI of 19 sub-industries in high-polluting industry.

#### 4.3.1. Malmquist Index and Trend Analysis

From the results of the MI of the heavy pollution industry as a whole (Table 5), China’s heavy pollution industry’s MI first shows an upward trend, which then changes to a downward trend. There was a negative growth in 2012–2013, 2013–2014, and 2014–1015. In 2012–2013, the EFF of heavy pollution industry did not change significantly, but the TECH of the industry has declined, and the technology of the industry has degraded by 1.5%. In 2013–2015, the EFF of heavy pollution industries declined, which led to a decline in the MI. In 2014–2015, the EFF and TECH of the industry declined, which led to a decrease in the overall MI. In the remaining years, the MI has been growing positively, especially the growth rate of 30.2% from 2004 to 2005. This is also related to the improvement of frontier effects, as well as the fact that the technological level of the entire industry is constantly improving. This clearly shows that the reason for the development of environmental efficiency in high-polluting industry is the change in technological capabilities.

From the perspective of specific sub-industries (Table 6), the difference in MI is relatively small, and there is a phenomenon of higher MI in industries with low environmental efficiency. The average value of MI of the three environmental efficiency groups exceeds 1, respectively, which are 1.103, 1.167, and 1.128. Among them, the highest was the Mining and Processing of Non-Ferrous Metal Ores (B09), with an average annual growth rate of 17.90%. This is mainly due to the high frontier effect of the industry, with an average annual growth rate of 11.11%. At the same time, the catch-up effect also increased by 6.14%. From 2001 to 2015, the MI of China’s heavy pollution industry was 1.127, the EFF increased by 1.9%, and the average annual TECH increased by 10.7%. The improvement of the environmental efficiency of heavy pollution industries come more from technological progress.

From the perspective of changes in the MI of the industry groupings (Figure 3), the overall trend of China’s heavy pollution industry from 2001 to 2015 showed a cyclical fluctuation that first increased and then decreased. This trend change reflects China’s policy towards heavy pollution industries, from high pollution, high energy consumption, large emissions to low pollution, low energy consumption, and small emissions, and from environmental damage to new energy and new technologies to the development of a green economy. At the same time, it can be seen that since 2011, the green development of high-polluting industry has gradually stabilized.

#### 4.3.2. Catch-Up Effect and Frontier Effect Index and Change Analysis

In this section, we divide each industry into three groups according to the environmental efficiency and analyze the efficiency changes of the sub-industries.

From the perspective of the efficiency change of the catch-up effect (Table 6), the higher the environmental efficiency of the industry, the smaller the catch-up effect index, and the smaller the role it plays. Among them, the low environmental efficiency group has a higher catch-up effect. For example, the Mining and Washing of Coal (B06) ranks third at 1.060, with an average annual growth rate of 5.96%; high environmental efficiency groups such as the Processing of Petroleum, Coking, and Processing of Nuclear Fuel (C25) have the lowest catch-up effect index of 0.974, with an average annual growth rate of −2.6%. This shows that the improvement of EFF has the greatest impact on industries with low environmental efficiency, while it has less effect on industries that already have a higher environmental efficiency.

From the perspective of the frontier effects of sub-industries (Table 6), the frontier effects of all heavy pollution industries are greater than 1, and they are all on an upward trend. Among them, Extraction of Petroleum and Natural Gas (B07) and Manufacture of Medicines (C27) in the high environmental efficiency group have changed significantly, with 14.63% and 11.71%, respectively. The environmental efficiency is relatively low in the Manufacture of Non-metallic Mineral Products (C30) and the Smelting and Pressing of Ferrous Metals (C31), both at 8.53%. This shows that the key to environmental efficiency is the change in technical efficiency, which results from the innovation capability of the industry. At the same time, technical capabilities have an obvious effect on the industries in high and medium environmental efficiency groups.

By observing the catch-up effect and frontier effect change trends of sub-industries (Figure 4 and Figure 5), We find that over the period from 2001 to 2015, the trend of catch-up effect had been relatively stable since 2009 and had been maintained at around 1. The changing trend of the frontier effect is basically the same as that of MI. There is a periodic characteristic of rising first and then falling, and the changes are relatively large. In addition, from 2008 to 2009, the frontier effect increased significantly, but the catch-up effect declined, indicating that the catch-up effect hindered the improvement of environmental efficiency in heavy pollution industries. At the same time, the various industries of the low environmental efficiency group are lower than the average level of the high-pollution industries in terms of catch-up effect and frontier effect. It shows that, out of all high-polluting industries, more focus should be placed on the group of low environmental efficiency.

## 5. Conclusions and Suggestions

This article examines the environmental efficiency of 19 heavy pollution sub-industries in China from 2001 to 2015 and uses the DEA model that includes undesirable output and MI in order to analyze the topic from both the static and dynamic perspectives.

From a static analysis of environmental efficiency, we obtained that the overall environmental efficiency of China’s heavy pollution industry has generally maintained an upward trend, but the overall environmental efficiency has not reached 1. The environmental efficiency of China’s highly polluting industries is still not at its best. Wu et al. (2019) conducted a study on 38 industrial sectors in China and confirmed that the environmental efficiency of 38 industrial sectors has been increasing from 2007 to 2011 as well, which is consistent with the results of this paper [36]. Xiao et al. (2018) conducted a study on the energy-environmental efficiency of China’s industry from 1995 to 2009, and the overall trend is also increasing [37]. Since 2001, from an environmentally friendly strategy to a beautiful China, China has continuously introduced various environmental protection policies, resulting in an overall increase in the environmental efficiency of high-polluting industries [38]. From the perspective of sub-industries, the environmental efficiency has significant inter-industry differences across China’s heavy pollution sub-industries. Xie et al. (2019) conducted a study on the environmental efficiency of 36 industries in China from 2006 to 2015 [39]. The study also showed that there are significant differences in environmental efficiency between industries. Based on this finding, this article divides 19 industries into three groups of high, medium, and low environmental efficiency for an in-depth analysis. Among them, the high environmental efficiency group is high-input, low-pollution industries, such as the Manufacture of Textile, Wearing Apparel and Accessories (C18); Manufacture of Leather, Fur, Feather and Related Products and Footwear (C19); etc.; the low environmental efficiency group is high-input and high-polluting industries, such as the Manufacture of Non-metallic Mineral Products (C30), the Mining and Washing of Coal (B06), etc. This classification is consistent with the results obtained by Kong (2020) when studying the environmental efficiency of China’s manufacturing industry and its industry heterogeneity [40]. It also classifies the Manufacture of Raw Chemical Materials and Chemical Products (C26), the Smelting and Pressing of Ferrous Metals (C31), the Manufacture of Paper and Paper Products (C22), and the Manufacture of Non-metallic Mineral Products (C30) as low-efficiency industries. Wu et al. (2019) measured the energy and environmental efficiency of China’s industrial sector [36]. His results showed that the environmental efficiency of Manufacture of Raw Chemical Materials and Chemical Products (C26) is low, but the difference between that study and this article is that the results of that paper illustrate that the Processing of Petroleum, Coking, and Processing of Nuclear Fuel (C25) is inefficient. This difference may be due to the fact that his article uses a non-homogeneous input–output DEA model and different indexes.

From the perspective of dynamic analysis of environmental efficiency, the overall trend of environmental efficiency follows a regular pattern of a periodic fluctuation that first increases and then decreases. This trend change reflects China’s policy towards heavy pollution industries, from high pollution, high energy consumption, large emissions to low pollution, low energy consumption, and small emissions, and from environmental damage to new energy and new technologies to the development of a green economy. The change in the catch-up effect has become flat since 2009, and the trend of the frontier effect is basically consistent with MI, which shows that the improvement of environmental efficiency depends on the improvement of technological capabilities. In addition, the MI is relatively close, but there is a phenomenon of low environmental efficiency and high MI. This is because high-polluting industries with lower environmental efficiency have greater potential for environmental improvement, and therefore, under the premise of the country’s continuous environmental policies, environmental efficiency has improved at a faster rate.

In view of the obtained results, this article puts forward the following countermeasures and suggestions:

Reduce energy input in high-polluting industry. From 2001 to 2015, the overall environmental efficiency of China’s heavy pollution industry did not reach 1, which means that environmental efficiency has not reached the optimal level in 15 years. In order to improve the environmental efficiency of high-polluting industry, on the one hand, the authorities should implement corresponding policies to limit energy input in various industries to reduce energy consumption and, on the other hand, increase capital and labor input in various industries. In this way, the technical level of various industries may be increased, thereby improving environmental efficiency. In particular, the Smelting and Pressing of Ferrous Metals (C31), which has low environmental efficiency, requires a lot of energy input and needs to limit the energy consumption of the industry.

Reduce the volumes of the three main categories of emissions (waste gas, wastewater, and solid waste). The policymakers can formulate various rules and regulations in order to limit the emissions of different form, minimizing undesirable output. The Production and Supply of Electric Power and Heat Power (D44), the Manufacture of Paper and Paper Products (C22), and the Mining and Washing of Coal (B06) rank first in the emissions of waste gas, wastewater, and solid waste. It is necessary to increase efforts to cut down the emissions of exhaust gas, wastewater, and solid waste in these three industries.

There are obvious differences in environmental efficiency between high-polluting industries, so the government can formulate different policies for different industries. According to the classification of high-pollution industries in this article, more attention should be paid to the low environmental efficiency group, and the low-efficiency group should learn from the high environmental efficiency group to enhance energy conservation and emission reduction.

The annual average EFF of the high environmental efficiency group in the heavy pollution industry is less than 1, indicating that the high pollution industries in the high environmental efficiency group may have problems with production management and other aspects, which need to be resolved. The EFF and TECH of the low environmental efficiency group are lower than the average level of the high-polluting industries. The technical capabilities of these industries should be further strengthened. The government should increase capital and labor input to improve the technical capabilities of the industry.

## Figures and Tables

**Figure 1 ijerph-18-05761-f001:**
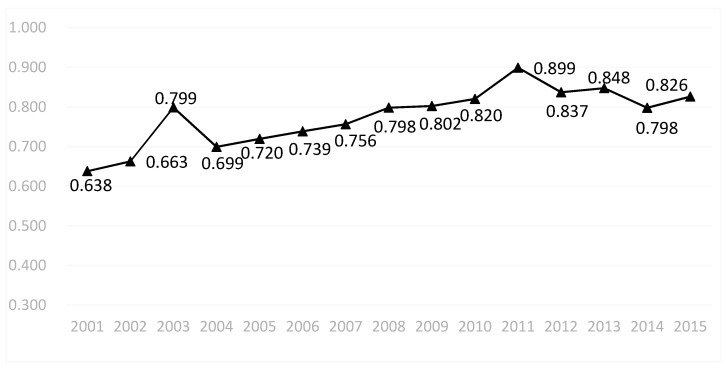
Changes in environmental efficiency of high-polluting industry from 2001 to 2015 in China.

**Figure 2 ijerph-18-05761-f002:**
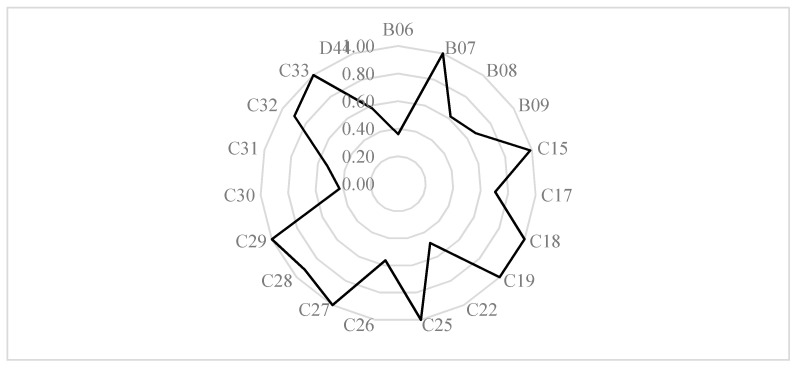
The average environmental efficiency of high-polluting sub-industries.

**Figure 3 ijerph-18-05761-f003:**
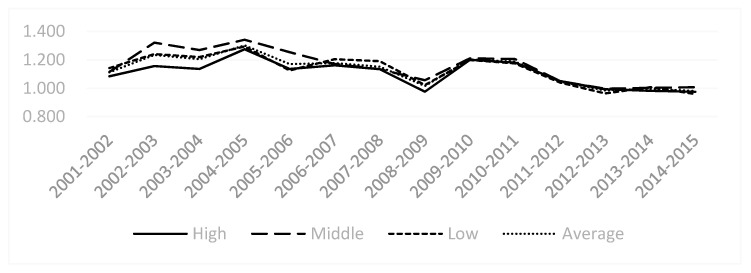
Malmquist index of China’s high-polluting industry 2001–2015.

**Figure 4 ijerph-18-05761-f004:**
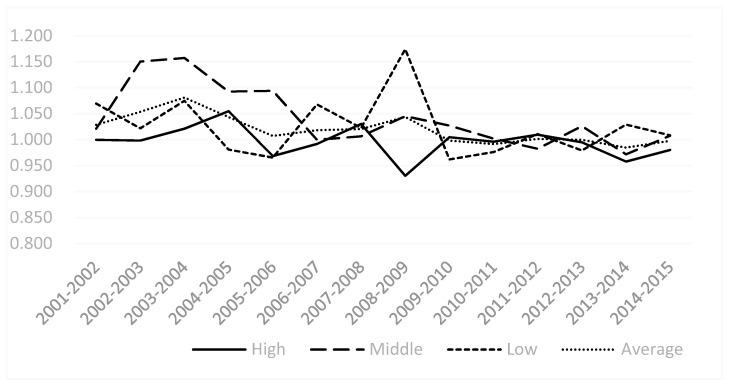
The catch-up effect of China’s heavy pollution industry from 2001 to 2015.

**Figure 5 ijerph-18-05761-f005:**
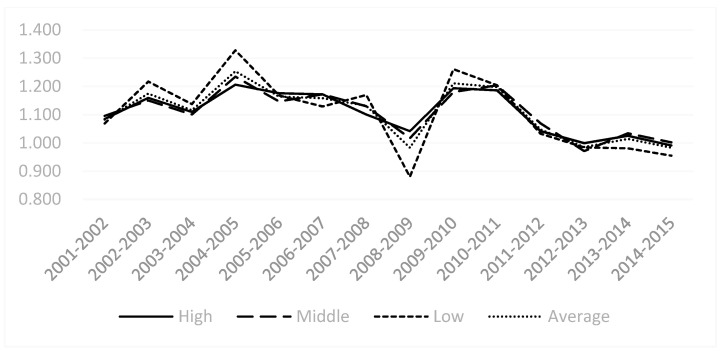
Frontier effects of China’s heavy pollution industry from 2001 to 2015.

**Table 1 ijerph-18-05761-t001:** Descriptive statistics of environmental efficiency evaluation indicators.

Indexes			N	Min	Max	Mean	Standard Deviation
Input indexes		Comprehensive energy consumption (E)	285	217.60	69,342.42	9722.82	13,471.15
		annual average number of employees (L)	285	24.37	652.06	241.18	164.51
		the net fixed assets (C)	285	110.4	74,494.94	6634.26	10,174.45
Output indexes	Desirable output	main business income (I)	285	178.54	83,564.54	16,586.61	17,394.61
	Undesirable output	wastewater discharge (W)	285	3724.00	424,597.00	87,860.00	99,818.45
		exhaust gas discharge (G)	285	119.00	225,447.00	21,486.10	44,295.06
		solid waste discharge (S)	285	0.00	619.00	43.25	92.80

**Table 2 ijerph-18-05761-t002:** The main data of the indicators of 19 high-pollution sub-industries.

Industry	Input Indexes			Output Indexes			
				Desirable Output	Undesirable Output		
	Comprehensive Energy Consumption	Annual Average Number of Employees	The Net Fixed Assets	Main Business Income	Wastewater Discharge	Exhaust Gas Discharge	Solid Waste Discharge
B06	9258.86	463.84	7756.84	16,104.90	90,322.20	2125.81	281.32
B07	4048.97	86.16	7901.38	8030.21	10,344.20	1106.69	1.11
B08	1309.52	51.00	1171.85	4161.91	17,101.60	1864.21	103.75
B09	875.28	49.12	855.74	3059.66	39,491.67	620.85	123.51
C15	1140.35	117.92	2278.40	7819.24	58,747.00	1732.85	4.24
C17	5753.66	554.52	5189.89	21,585.97	195,976.53	3029.13	5.26
C18	687.69	387.33	1623.90	10,315.47	12,887.13	241.07	0.37
C19	407.26	242.74	883.64	6733.85	20,608.47	278.81	0.44
C22	3616.54	136.18	3089.79	7639.16	343,278.07	6134.15	10.34
C25	14,535.07	80.98	5100.68	22,405.61	69,537.00	14,223.19	38.93
C26	30,103.96	406.31	12,487.87	39,013.73	302,891.07	23,719.17	37.26
C27	1463.04	156.36	2816.14	10,277.77	46,407.87	1908.18	2.12
C28	1642.98	43.30	1359.89	4169.99	45,569.53	2661.06	1.77
C29	3079.08	296.54	3615.71	15,117.05	10,872.33	2042.61	6.46
C30	25,088.25	482.43	8689.09	25,505.22	36,205.40	80,011.65	44.97
C31	47,911.09	321.40	13,493.64	41,030.80	141,051.13	103,341.23	88.78
C32	11,198.76	161.57	5410.22	23,411.91	32,030.73	21,727.91	24.27
C33	3053.99	283.38	3346.63	16,590.91	25,962.67	2966.74	3.40
D44	19,559.18	261.27	38,979.63	32,172.27	170,055.33	138,500.55	43.51

**Table 3 ijerph-18-05761-t003:** 2005–2015 Environmental efficiency of various heavy pollution sub-industries in China.

	2001	2002	2003	2004	2005	2006	2007	2008	2009	2010	2011	2012	2013	2014	2015
B06	0.124	0.148	0.164	0.210	0.269	0.276	0.291	0.379	0.387	0.450	1.000	0.581	0.402	0.407	0.320
B07	1.000	1.000	1.000	1.000	1.000	1.000	1.000	1.000	1.000	1.000	1.000	1.000	1.000	1.000	1.000
B08	0.236	0.250	0.326	0.346	0.427	0.445	0.502	0.659	0.721	1.000	1.000	1.000	1.000	0.765	0.594
B09	0.277	0.289	0.338	0.425	0.522	0.667	0.601	0.631	1.000	1.000	1.000	0.663	1.000	0.655	1.000
C15	1.000	1.000	1.000	0.815	1.000	1.000	1.000	1.000	1.000	1.000	1.000	1.000	1.000	1.000	1.000
C17	0.396	0.507	1.000	0.488	0.549	0.585	0.571	0.529	0.585	0.665	1.000	0.702	1.000	1.000	1.000
C18	1.000	1.000	1.000	1.000	1.000	1.000	1.000	1.000	1.000	1.000	1.000	1.000	1.000	1.000	1.000
C19	1.000	1.000	1.000	1.000	1.000	1.000	1.000	1.000	1.000	1.000	1.000	1.000	1.000	1.000	1.000
C22	0.398	0.424	1.000	0.389	0.441	0.445	0.451	0.462	0.452	0.487	0.510	0.479	0.452	0.451	0.469
C25	1.000	1.000	1.000	1.000	1.000	1.000	1.000	1.000	1.000	1.000	1.000	1.000	1.000	1.000	1.000
C26	0.445	0.477	0.695	0.531	0.463	0.461	0.564	0.688	0.568	0.544	0.572	0.567	0.551	0.549	0.760
C27	1.000	1.000	1.000	1.000	1.000	1.000	1.000	1.000	1.000	1.000	1.000	1.000	1.000	1.000	1.000
C28	0.534	0.654	1.000	1.000	1.000	1.000	1.000	1.000	1.000	1.000	1.000	1.000	0.825	0.779	1.000
C29	1.000	1.000	1.000	1.000	1.000	1.000	1.000	1.000	1.000	1.000	1.000	1.000	1.000	1.000	1.000
C30	0.272	0.298	0.330	0.295	0.271	0.291	0.318	0.404	0.471	0.478	0.485	0.437	1.000	0.519	0.535
C31	0.342	0.366	0.689	0.530	0.462	0.412	0.604	1.000	0.525	0.552	0.515	0.475	0.453	0.529	0.501
C32	0.521	0.547	1.000	0.665	0.743	1.000	1.000	1.000	1.000	1.000	1.000	1.000	1.000	1.000	1.000
C33	1.000	1.000	1.000	1.000	1.000	1.000	1.000	1.000	1.000	1.000	1.000	1.000	1.000	1.000	1.000
D44	0.579	0.631	0.633	0.593	0.528	0.451	0.469	0.413	0.535	0.407	1.000	1.000	0.420	0.506	0.518

**Table 4 ijerph-18-05761-t004:** Grouping of high-pollution sub-industries in China.

Group	Specific Industry	Mean Environmental Efficiency
High Environmental Efficiency Group	Extraction of Petroleum and Natural Gas (B07); Manufacture of Textile, Wearing Apparel and Accessories (C18); Manufacture of Leather, Fur, Feather and Related Products and Footwear (C19); Processing of Petroleum, Coking and Processing of Nuclear Fuel (C25); Manufacture of Medicines (C27); Manufacture of Rubber and Plastics Products (C29); Manufacture of Metal Products (C33)	1.000
Middle Environmental Efficiency Group	Manufacture of Liquor, Beverages and Refined Tea (C15); Manufacture of Chemical Fibres (C28); Smelting and Pressing of Non-ferrous Metals (C32); Manufacture of Textile (C17); Mining and Processing of Non-Ferrous Metal Ores (B09); Mining and Processing of Ferrous Metal Ores (B08)	0.800
Low Environmental Efficiency Group	Production and Supply of Electric Power and Heat Power (D44); Manufacture of Raw Chemical Materials and Chemical Products (C26); Smelting and Pressing of Ferrous Metals (C31); Manufacture of Paper and Paper Products (C22); Manufacture of Non-metallic Mineral Products (C30); Mining and Washing of Coal (B06)	0.491

**Table 5 ijerph-18-05761-t005:** 2001–2015 Malmquist index of high-polluting industry and its breakdown.

Year	Catch-Up	Frontier	MI
2001–2002	1.028	1.083	1.112
2002–2003	1.054	1.175	1.235
2003–2004	1.081	1.116	1.205
2004–2005	1.043	1.254	1.302
2005–2006	1.007	1.164	1.170
2006–2007	1.018	1.158	1.178
2007–2008	1.020	1.132	1.152
2008–2009	1.044	0.983	1.016
2009–2010	0.998	1.211	1.204
2010–2011	0.992	1.198	1.187
2011–2012	1.001	1.047	1.047
2012–2013	1.000	0.985	0.985
2013–2014	0.985	1.014	0.997
2014–2015	0.998	0.983	0.981

**Table 6 ijerph-18-05761-t006:** MI decomposition results of heavy pollution sub-industries.

Industry	Catch-Up	Frontier	Malmquist
B07	0.982	1.146	1.140
C18	1.008	1.106	1.112
C19	1.000	1.074	1.075
C25	0.974	1.101	1.073
C27	1.000	1.117	1.117
C29	1.000	1.111	1.111
C33	1.004	1.089	1.094
High Environmental Efficiency Group	0.996	1.106	1.103
C15	0.999	1.129	1.127
C28	1.041	1.107	1.155
C32	1.050	1.110	1.156
C17	1.033	1.097	1.133
B09	1.061	1.111	1.179
B08	1.064	1.093	1.167
Middle Environmental Efficiency Group	1.042	1.108	1.153
D44	1.003	1.184	1.167
C26	1.019	1.089	1.100
C31	1.018	1.085	1.101
C22	1.008	1.113	1.121
C30	1.040	1.085	1.115
B06	1.060	1.093	1.161
Low Environmental Efficiency Group	1.024	1.108	1.128
Average	1.019	1.107	1.127

## Data Availability

Not applicable.

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
