# Peer review of "Environmental Efficiency Assessment of Heavy Pollution Industry by Data Envelopment Analysis and Malmquist Index Analysis: Empirical Evidence from China"

_ijerph, 2021, doi:10.3390/ijerph18115761_

Round 1
Reviewer 1 Report
Summary
In this study, the environmental efficiency of heavy pollution industry in China is investigated using data envelopment analysis including pollution produced (solid, waste water and gaseous), from 2001 to 2015. The Malmquist Index is calculated to determine the change in environmental efficiency with time. Overall, environmental efficiency is shown to be improving slowly over time. The 19 sub-industries were divided into three groups of high, medium and low environmental efficiency. The high environmental efficiency group comprises high-input, low-polluting industries. The low environmental efficiency group consists of high-input, high-polluting industries. Industries with low environmental efficiency generally have higher Malmquist Index. It is concluded that environmental efficiency improvements depend on improvements in technological capabilities, and the focus should be on reducing energy inputs, reducing polluting emissions and maximizing production.
I find this manuscript to be well structured and clearly presented.
Major Issues
There are no major issues to be addressed.
Minor Issues
Please address the following:
- Abstract: There is tautology in the following sentence: ‘The general trend shows a regular pattern of a periodic fluctuation that first increases and then decreases’. Please rephrase.
- The opening sentence of the introduction (‘Due to economic globalization, the economies of all countries in the world are constantly developing’) is a generalization that may not hold in all countries (e.g. countries where there is a war). Please rephrase.
- First paragraph of introduction: It is not obvious that the industrial sector drives economic development in most developing countries, as implied here. Please either support this statement with a reference or rephrase.
- Section 3.1.1: Please reference Charnes and Cooper (and Rhodes?) as the primary source.
- Section 3.1.1 first paragraph: it is odd to describe ‘waste water, waste gas, and solid waste emissions’ as ‘environmental inputs’. While they are inputs to the model, they are outputs of the production process.
- Section 3.1.1 first paragraph: References need to be provided on ‘reciprocal method, hyperbolic processing method, and linear transformation method.’
- Section 3.1.1 second paragraph: References are needed for the slack based model.
- Page 5 section 3.2: Several inputs with environmental implications have been omitted from the study, including input of raw materials (water, chemicals e.g. catalysts and raw materials that are transformed through production). It needs to be motivated why these inputs are not considered. Also, what about land area occupied for the industrial processes?
- Section 3.2 last line: What exhaust gases are considered? In particular, are greenhouse gas emissions considered, or just air pollutants? Are the different gases weighted in any way? I assume fugitive emissions (e.g. dust emissions from mining) are not considered? Please clarify.
Reviewer 2 Report
Dear Authors,
The manuscript is interesting and up-to-date. This study represents a contribution to the current debate about the environment and pollution. Aspects of environmental efficiency play an essential role in the debate but below, I present several suggestions for improvements:
1. My general comment is that the authors have used several times the term "environmental efficiency" without providing a clear definition of it. It is necessary to explain how the environmental efficiency was defined and scored. Additionally, the authors use the term "environmental performance" without providing a clear definition/distinction from, e.g., "environmental efficiency."
2. What was the goal of the paper? It is not clear. The Authors must add a clear statement about it. The phrase "comprehensive analysis" is not a precise aim. Readers should know why you would like to analyze, what you would like to prove or find.
3. I suggest improving this part: ( page 2. "How to evaluate the impact of heavy waste, we should consider the energy inputs in heavy pollution industry as well, thus, addressing both sides of the production process. Therefore, it is crucial to measure the environmental efficiency of the heavy pollution industry"). The explanation expressed is unclear, hard to follow, and lacks a logical sequence.
Authors should improve the complete introduction to give the reader a broader contest of the problem, explaining why it is crucial to evaluate the environmental efficiency in China as you offer. What proposed approach will bring, what is an additional value comparing to existing knowledge? It is different, but different doesn't necessarily mean "better," more valuable.
Some data about production in China would make the need for such analysis easier to present and they might be helpful in the context of results and conclusions. Without knowledge about China, it is impossible to explain the values of environmental efficiency presented in the article.
3. The method for calculating efficiency data should be capitalized as "Data Envelopment Analysis" because it is the method's name (page 2).
4. The Table 1 title suggests that the data presented will be from 2005 to 2015 (Table 1. 2005-2015 ), but the data are from 2001-2015. I suggest you add "in China" in the title of table 1 and graph 1.
5. Under table 1 and Figure 2, an explanation of B06, B07 ... D44 etc., is needed.
6. Please add an explanation of the grouping of high-pollution sub-industries in China. Why in three groups? What were the criteria of grouping? (page 9)
7. The Authors use the abbreviation: "The annual average EFF" without its explanation. page 13.
8. No discussion with other articles provided.
9. In the results and conclusions you have mentioned that "environmental efficiency follows a regular pattern of a periodic fluctuation that first increases and then decreases" but there is no explanation why, is this a trend described by others? etc. And some other results are without any explanation.
Reviewer 3 Report
This article addresses an important topic using a novel method, combining DEA and MI to quantitively measure the environmental efficiency of multiple heavy-polluting industries in China. The logic is clear. The language used is straightforward. The structure is complete. The unique contribution of this article to the field is also clearly stated in Literature Review part. The authors explain some economic concepts/terminologies in a way that are easy for general public readers or environmental professionals to understand, which is very important. Another important contribution is to present and compare various sub-industries in China. Besides analytical results, the article further suggest practical policy-oriented actions. I don’t have much comments except a few minor ones:
- Introduction, second paragraph: I would suggest avoid using subjective word like “superior” in a scientific article. Use “more advantage” or specifically what is “superior”, faster? More accurate? etc.
- Introduction, third paragraph: although I understand what authors mean, dispute can be avoided by just say China is a representative developing economy or one of the most representative developing economies in the world. It is not the point to argue which country is “the most representative”, a rather subjective point, in this scientific article.
- 3.1.1, first paragraph, “backup” is recommended to add to the end of the last sentence.
- 3.1.1, please keep all variables font consistent throughout
- 3.1.1, it is recommended to directly point out the definition of environmental efficiency in SBM model
- 4.1.1, first paragraph, please have the software version and distributor
Reviewer 4 Report
This paper uses DEA and Malmquist index for assessing the environmental efficiency of the heavy pollution industry in China. The authors use data from 2001 to 2015. The idea is interesting. However, the method lacks of adequate robustness and the results are inadequate for drawing appropriate conclusions and recommendations. Authors have to redo their analyses using a robust approach. If they want to stick with DEA and MI, then they should bootstrap both approaches.
Issues
- The papers requires serious proofreading.
- “but it has not reached the state of full effectiveness yet.” This is an awkward comment. It is not expected that the industry will reach full effectiveness. In fact, DEA and related approaches are known for estimating the “relative” efficiency, which means is relative to the top performers.
- Page 1. Hasanov et al. 2019 is cited in the paper but it is not listed in the references section.
- Page 2. Park et al. (2018) is cited in the paper but it is not listed in the references section.
- Page 3. Twum et al. (2020) is cited in the paper but it is not listed in the references section.
- Page 3. It says “Li et al. (2020)”, it should be “Li and Xiao (2020)”.
- Page 4. “ However, the DEA model also has certain shortcomings. It can only evaluate the environmental efficiency of the industry from a static perspective.” This is an incorrect affirmation. DEA can perform dynamic evaluations, it depends how it is used by researchers.
- Page 4. “It is believed that the way to increase efficiency is to reduce input in addition to expanding output, but expanded output also includes undesired output, which is contrary to the purpose of improving efficiency.” Who believes this? It is not a matter of believes. It is a matter of a mathematical relationship. The mathematical formulation of DEA dictates the possible ways for increasing efficiency.
- Across the article. Do not use “expected” output and “unexpected” output. The common terms in the DEA literature are desirable and undesirable outputs, respectively.
- When introducing DEA and the Malmquist index you should cite the seminal papers.
- The authors have to show the statistics summary of each one of 19 considered sub-industries. It looks, based on the results from table 1, that such industries are not comparable. That means that maybe they have to be treated independently or grouped by a data mining technique.
- Table 1 shows serious issues with this research. First, there are 7 out of 19 subindustries that are absolutely efficient. This arises the concern about the comparability of these subindustries. However, it is hard to say because there is not any information about inputs and outputs. Second, there are a lot of 1’s in table 1, which confirms the poor discrimination power of the classic DEA approach. The authors should do a bootstrap procedure for estimating the bias-corrected efficiency scores (proving that the subindustries are in fact comparable). Third, the table suggest an unclean set of data. For example, B06 efficiency for year 2011 clearly is an outlier. Similarly, C30 year 2013.
- Another major problem is the average of the environmental efficiency. Again, without knowing the data set, it looks this is a biased approach, where the bunch of efficient subindustries (discussed above) are inflating the efficiency. Again, a bootstrap procedure should be utilized before estimating the average (proving that the subindustries are comparable).
- Was the data set normalized? The results suggest a high variability in the data set.
- The Malmquist index has to be bootstrapped as well. Otherwise uses biased efficiency estimations.
- Authors recommend decrease energy consumption and reduce emissions. These are general pretty intuitive recommendations. I do not need to do a DEA or Malmquist index approach for identifying such recommendations. The authors have to work harder on the interpretation of results. What about target values? Authors could estimate the target values for such reductions and provide stronger recommendations.
Round 2
Reviewer 2 Report
Dear Authors,
Thank you for all your efforts concerning improvements. I am accepting almost all of them, but I would like you to consider again my previous comment"Some data about production in China would make the need for such analysis easier to present and they might be helpful in the context of results and conclusions. Without knowledge about China, it is impossible to explain the values of environmental efficiency presented in the article."
My intention was to encourage you to present (w.g., in a table) some (main) data about the industry in China including Heavy Pollution Industry.
Best regards
Reviewer 4 Report
The authors want to stick with no robust methods of analysis which I believe is a pity, this could be a nice research article with the appropriate method of analysis.
Major issues
- The authors have to add a table with the statistical summary (mean, standard deviation, max and min values, etc) of inputs and outputs. They have to indicate the direction of each input and output (desirable, undesirable). They have to discuss if there is or there is not any significant correlation between inputs and outputs. Furthermore, how the authors support the selection of inputs and outputs?
- They did not respond if the data set was normalized before running it.
- Are the authors using the aggregate value of inputs and outputs for each subindustry? Or have the authors the information of inputs and outputs of each company from each subindustry? If the authors are using aggregate data, then they have to support this decision.
- How were the ranges for table 2 selected? They look like an arbitrary decision.
